# Rare Case of a Peripheral Giant Cell Granuloma of the Jaw as First Manifestation of Primary Hyperparathyroidism

**DOI:** 10.3390/diagnostics12123018

**Published:** 2022-12-02

**Authors:** Samanta Buchholzer, Tommaso Lombardi

**Affiliations:** Oral Medicine and Oral Maxillo-Facial Pathology Unit, Division of Oral and Maxillofacial Surgery, Department of Surgery, Faculty of Medicine, University Hospitals of Geneva, University of Geneva, 1205 Geneva, Switzerland

**Keywords:** peripheral giant cell granuloma, central giant cell granuloma, hyperparathyroidism, benign tumors

## Abstract

Giant cell granulomas (GCG) are uncommon benign tumor-like lesions mostly arising in the oro-facial area. They are more common in women and occur in patients younger than 30 years. Lesions restricted to the bone are referred to as central giant cell granulomas (CGCG), and those developing primarily on soft tissues are termed peripheral giant cell granulomas (PGCG). Both types are histologically identical. The combination of both clinical examination and radiography allows for the differentiation of those two variants. On rare occasions GCG, and especially CGCG, may develop in relation to hypercalcemia linked to hyperparathyroidism (HPT). In those cases, the GCG treatment prognosis is closely linked to the HPT management. Therefore, patients diagnosed with a GCG must be investigated to search for an HPT. Reported herein is a rare clinical case of a mandibular PGCG which led to the diagnosis of primary HPT.

**Figure 1 diagnostics-12-03018-f001:**
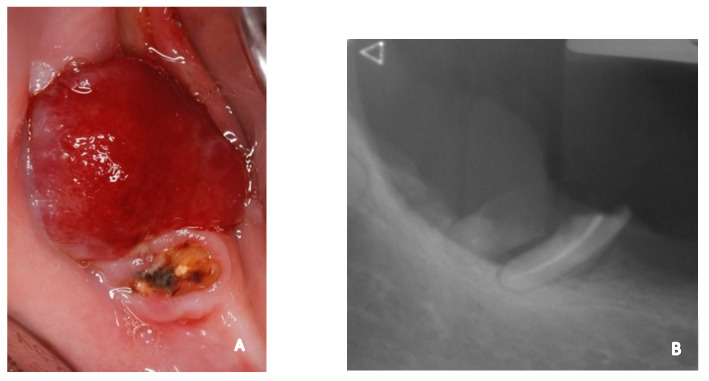
Intra-oral initial occlusal view showing an erythematous nodule on the posterior right mandibular crest, overhanging the distal root remain of tooth 47 (**A**). Retro-alveolar radiograph showing a bone lysis involving the root remains of tooth 47 (**B**). A 57-year-old Hispanic woman presented herself at the emergency department of the Oral and Maxillo-facial Department at the Geneva University Hospital because of the development of an intra-oral lump localized in the fourth quadrant within the duration of one month. The general anamnesis revealed numerous past episodes of renal colic in relation to kidney stones. She had a daily prescription of an angiotensin II receptor antagonist to treat arterial hypertension and a serotonin uptake inhibitor to treat depression. The intra-oral inspection revealed a polypoid erythematous nodule measuring 1.5 × 1.4 × 1 cm, with an irregular surface, positioned on the posterior mandibular crest in region 47 (**A**). The lesion overhung the distal remnant root of tooth 47. Palpation induced bleeding from the nodule and revealed an elastic feature. A retro-alveolar radiography confirmed the presence of the root remnants of tooth 47 with a cervical resorption of the distal one and endodontic treatment of the mesial one, both surrounded by severe alveolar bone loss. However, the mandibular bone frame remained within normal limits (**B**).

**Figure 2 diagnostics-12-03018-f002:**
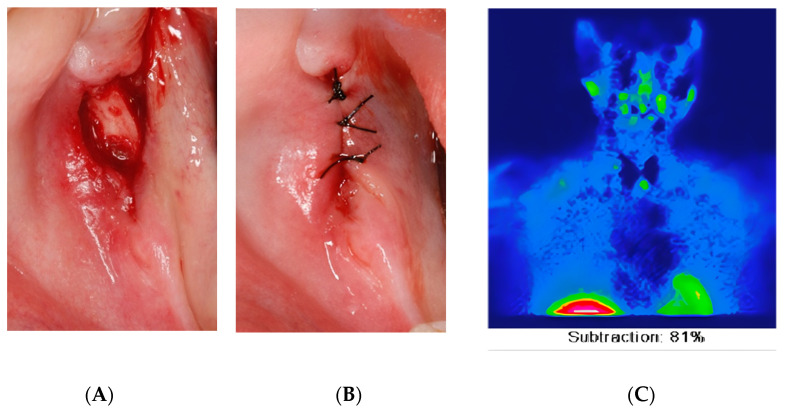
Intra-oral clinical occlusal view showing the mandibular bone crest after the nodule excision (**A**) and the wound closure after the nodule excision (**B**). Tc04/Tc99m–MIBI Scintigraphy (**C**). An excisional biopsy of the lesion and the removal of the remnant roots of tooth 47 were completed (**A**,**B**) and confirmed the diagnosis of giant cell granuloma. In order to rule out a brown tumor, a blood analysis was requested and revealed elevated serum calcium and parathyroid hormone (PTH) levels and low serum phosphate levels compared to the normal values. The patient was then referred to the Endocrinology department for further complete complementary assessment and the search for hyperparathyroidism. The biological analysis revealed an elevated corrected calcium level at 2.78 mmol/L (normal value: 2.2–2.52 mmol/L), an elevated PTH level at 14.2 pmol/L (normal value 1.1–6.8 pmol/L), a slightly reduced phosphate level at 0.57 mmol/L (normal value 0.8–1.45 mmol/L) and a vitamin D level at 34 nmol/L (normal value > 75 nmol/L). Additionally, blood beta-crosslaps were increased to 1′253 ng/L, indicating bone remodeling. As the patient met the criteria for surgical treatment, radiological investigations were undertaken. The thyroid ultrasound revealed a multinodular thyroid with nodular lesions in the left inferior part and two nodules in the right lobe. In addition, the parathyroid 99 technetium scintigraphy showed a suspicious hyperfixation of the left lower lobe, consistent with a parathyroid adenoma (Figure 2C). The complete excision of the inferior left parathyroid containing the lesion was undertaken under general anesthesia. The histopathological analysis confirmed the diagnosis of a parathyroid adenoma. The post-operative biological analysis revealed normal levels of corrected calcium and parathoromone, which is in agreement with the biological healing of the primary hyperparathyroidism. In order to avoid the hungry bone syndrome, the patient received calcium supplementation, and the close monitoring of serum calcium levels was conducted. No recurrence of HPT nor oral GCG were observed at 5 years follow-up, assuming the effective management of the patient’s condition.

**Figure 3 diagnostics-12-03018-f003:**
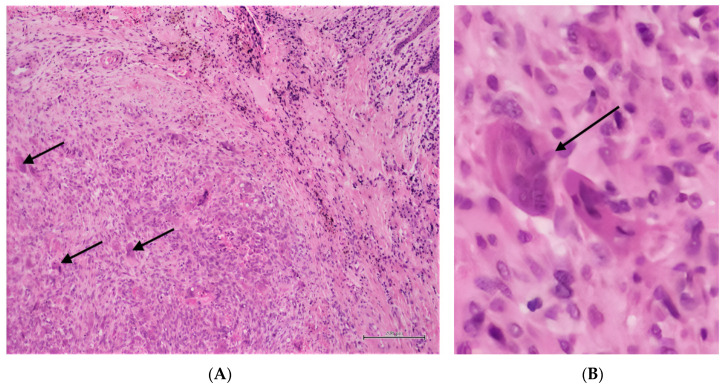
Histopathological analysis of the intra-oral lesion provided the diagnosis of giant cell granuloma ((**A**): H&E, ×10; (**B**): H&E, ×30). The presence of multinucleated giant cells (arrows) within a population of fibroblasts and some small vessels. The lesion is well circumscribed and separated by a band of fibrous connective tissue containing scattered lymphocytes and hemosiderin deposits. GCG, and especially PCGC, generally occur in response to chronic local irritation. CGCG may develop in association with genetic disorders such as Cherubism, type 1 Neurofibromatosis (NF1) and Noonan syndrome, thus, implying other specific features of each disease, as these lesions will have similar histopathological features as CGCG [1]. GCG, and especially CGCG, may be related to hyperparathyroidism (HPT) as a result of an abnormality of the bone metabolism linked to hypercalcemia. Similarly to isolated GCG, GCG linked to HPT occur mostly in postmenopausal women and are generally located in the posterior mandibula [2]. HPT remains asymptomatic in more than 75% of cases; it is mostly diagnosed by routine blood tests showing high PTH levels in the context of a normal, low or high calcium levels and low or high phosphate levels. In limited cases, HPT may be revealed by the systemic effects caused by hypercalcemia affecting possibly the neurologic, renal, skeletal, digestive and cardiac systems [3]. The clinical presentation of primary HPT was first described by Albright in 1930 as a “disease of stones, bones and groans” with florid manifestations including nephrocalcinosis and nephrolithiasis, brown tumors of the bones, muscle weakness and neurological conditions such as confusion, dementia, depression and memory loss. Since the advent of routine serum calcium measurement in the 1970s, most patients remain asymptomatic or pauci symptomatic with subclinical manifestations of mild hypercalcemia, such as osteoporosis, vertebral fractures and nephrolithiasis [3]. GCG are rarely the primary and only manifestation of HPT, and in those cases, CGCG referred as brown tumors are mostly concerned [2]. CGCG related to HPT are in the strong majority related to primary HPT which is caused in 80% of cases by parathyroid adenomas, in 10–15% of cases by parathyroid hyperplasia and in 1% of cases by parathyroid carcinoma. They are reported in 4.5% [2] to 5.9% [4] of patients with primary HPT. Brown tumors related to secondary HPT, linked to chronic kidney disease, are much rarer as they are only reported in 0.3% [3] to 1.7% of those patients [2]. Isolated GCG and GCG related to HPT have the same clinical, radiological and histological features; therefore, biological investigations of PTH, calcium and phosphate levels are necessary to diagnose or exclude an HPT. PGCG are rarely associated with HPT and often occur in the early stage of a primary HPT [4]. However, isolated PGCG remains a frequent oral lesion accounting for 10% of all gingival lesions [4], and HPT is not an uncommon disorder with a prevalence of 1–3% in the population [5]. The potential association of GCG with HPT should be acknowledged by dentists and oral specialists, and every patient diagnosed with GCG should benefit from a serum PTH, calcium and phosphate measurement in order to exclude a potentially associated HPT. Moreover, if the HPT is confirmed, the patient should be referred to its general physician or to an endocrinologist to search for possible HPT systemic manifestations and proceed with an adequate treatment. In case of suspicion of GCG and particularly PCGC, with or without association to HPT, the clinical examination should always be completed by a radiography in order to assess the extent of the lesion. The complete removal of the GCG is crucial in order to prevent its recurrence. We recommend to proceed with clinical controls until complete wound healing, and in case of CGCG, to follow-up with regular radiologic controls to ensure proper bone healing after the excision.

## Data Availability

Not applicable.

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
