# Peer review of "Rare Case of a Peripheral Giant Cell Granuloma of the Jaw as First Manifestation of Primary Hyperparathyroidism"

_diagnostics, 2022, doi:10.3390/diagnostics12123018_

Round 1
Reviewer 1 Report
The images are useful to report the rare case of peripheral giant cell granuloma linking with the diagnosis of primary hyperparathyroidism. One minor suggestion is that a H&E image of higher magnification, e.g. x100, can be added in Figure 3 so that readers can observe the histopathological changes clearly and easily.
Author Response
Dear Reviewer,
Thank you for your pertinent suggestion.
An additional H&E image of higher magnification was added in figure 3.
Reviewer 2 Report
An interesting case and presented well. My suggestion is to replace the H&E figure with a better quality and brighter image. If you have a scanner, it will be better to scan the slide and then capture a better image. Minor English corrections are also needed.
Author Response
Dear Reviewer,
Thank you for your pertinent remarks.
An additional H&E image of higher magnification was added in figure 1.
Also minor English corrections were made (see manuscript: text highlighted in yellow).